# Analysis of Educational Attainment in a Mexican Psychiatric Patient Population with Bipolar or Psychotic Disorders

**DOI:** 10.3390/brainsci13060881

**Published:** 2023-05-30

**Authors:** Hugo Cano-Ramirez, Lina Diaz-Castro, Kurt Leroy Hoffman

**Affiliations:** 1Center for Investigation in Reproduction, Autonomous University of Tlaxcala-CINVESTAV, Tlaxcala 90000, Mexico; 2Department of Epidemiological and Psychosocial Investigation, National Institute of Psychiatry Ramón de la Fuente Muñiz, Mexico City 14370, Mexico

**Keywords:** schizophrenia, bipolar, schizoaffective, educational attainment

## Abstract

Schizophrenia has been associated with premorbid poor educational performance and low educational attainment (EA). However, some studies have found positive associations between psychotic disorders and excellent scholastic performance. In the present study, we examined the association between EA and several clinical and nonclinical characteristics in psychiatric patients diagnosed with psychotic or bipolar disorders. Data were obtained from the files of 1132 patients who entered a major Mexico City psychiatric hospital during the years 2009–2010 for the treatment of psychotic symptoms and who were subsequently diagnosed with schizophrenia, bipolar, schizoaffective, or another psychotic disorder. Chi-squared tests, *t*-tests, and Cox regression analysis were applied to explore associations between EA and factors including gender, familial history of mental illness, premorbid personality characteristics, age of symptom onset, diagnosis, civil status, and current employment. Family history of mental illness decreased the hazard of having lower EA (B = −0.137, *p* = 0.025, ExpB = 0.872, 95% CI = 0.774–0.983), while a schizophrenia diagnosis independently increased it (B = 0.201, *p* = 0.004, ExpB = 1.223, 95% CI = 1.068–1.401). In male patients (but not in females), family history of mental illness was significantly associated with higher EA, while in female patients, premorbid schizoid-like personality characteristics were associated with lower EA. For both genders, lower EA was associated with having more children and being employed in manual labor, while higher EA was associated with professional employment. Conclusions: Compared with bipolar disorder, a schizophrenia diagnosis is associated with lower EA; however, familial history of mental illness and premorbid schizoid-like characteristics independently favor higher and lower EA in males and females, respectively. Since lower EA is generally associated with a lower economic status, special preventative attention should be given to students at high risk for schizophrenia, particularly those displaying a schizoid-like personality.

## 1. Introduction

Suffering from a major mental illness such as depression or schizophrenia (SCH) during childhood, adolescence, or early adulthood substantially increases lifetime medical costs, unemployment, and supplementary security income and reduces Educational Attainment (EA), earnings, and quality of life [1]. Mental illness also represents a substantial global economic burden with economic costs in some countries estimated to be up to 5.46% of the Gross National Product [2]. Hospitalization accounts for the largest proportion of direct costs of mental illness, while costs associated with unemployment, disability, income assistance, and premature mortality represent the largest proportions of overall costs [3]. However, it has been estimated that programs directed specifically toward increasing EA in affected individuals could significantly mitigate negative lifetime effects of the early onset of severe mental illness, particularly in the areas of lifetime earnings and the need for supplementary security income [1].

Perhaps not surprisingly, mental illness negatively impacts an individual’s academic performance. Although overt psychotic symptoms of SCH typically do not manifest until the late teens or early twenties, subtle developmental deviances in motor, cognitive, emotional, and behavioral domains may be observed years before overt disease onset [4]. Agnew-Blais and colleagues [5] observed that children who later developed SCH showed a lower IQ from ages 4 to 7 years compared with healthy individuals and with those who later developed affective psychosis (BD). In a Dutch twin study, van Oel and colleagues [6] observed underperformance at school that manifested around the time of secondary school in children who later developed SCH. MacCabe and colleagues [7] reported that a decline in verbal ability between 13 and 18 years is a predictor of later affective and nonaffective psychosis, while this decline was not observed in those who later developed BD. A meta-analysis by Dickson and colleagues [8] found that poorer academic and mathematics achievement by age 16 is associated with a later SCH diagnosis, and individuals with SCH are less likely to enter higher education. It has also been reported that persons with BD are less likely to complete university studies compared with healthy controls [9].

A number of studies have examined the relationship between genetic vulnerability to psychotic disorders and EA [10]. Perhaps surprisingly, both positive and negative associations between EA and genetic vulnerability to psychotic disorders have been reported. Le Hellard et al. [11] identified several gene loci that covary with both schizophrenia and EA. Notably, some of these loci showed positive relationships with both measures (that is, are associated with both SCH and high EA), while others were positively associated with SCH and negatively associated with EA. In another study, single nucleotide polymorphisms (SNPs) positively associated with EA were found to be associated with early brain development, cognitive performance, and intracranial volume, as well as with an increased risk of bipolar disorder [12]. Ohi et al. [13] reported that genetic factors correlated with a high childhood IQ and high EA are associated with BD, while those correlated with a low general cognitive ability are associated with SCH. In a separate study, a higher polygenic risk for bipolar disorder and schizophrenia was associated with high EA while, paradoxically, being associated with deficits in general cognitive measures, such as verbal–numerical reasoning [14]. Notably, the relative effect of a SCH or BD diagnosis on EA was found to differ among countries as a function of the country’s income level [15], with higher-income countries showing a greater educational gap between individuals with SCH versus healthy control subjects compared with lower-income countries.

Moreover, some studies suggest a positive relationship between superior academic performance and the risk of psychotic disorder. For example, in the study conducted by Isohanni and colleagues [4], mentioned above, excellent school performance at age 16 in males was also found to be a significant risk factor for a later SCH diagnosis. Karlsson and colleagues [16] analyzed national school records from Iceland and found that individuals with a high academic level, as well as their close relatives, showed an increased risk of psychosis. Additionally, MacCabe and colleagues [17] found a positive relationship between grade point average scores at age 16 and the adult incidence of bipolar disorder. Vreeker and colleagues [18] reported that patients with BD-I had higher Educational Attainment than patients with SCH as well as compared with healthy controls, even though the measured IQ of the BD-1 patients was significantly lower than that of healthy controls. However, Carter and colleagues [19] found that intellectual functioning did not have a predictive relationship with later psychosis, and Kendler and colleagues [20] reported that scholastic achievement (grade point average at 9th grade) was not significantly different between patients with BD and healthy controls, while this measure was significantly lower in patients with SCH.

### Aims of the Present Study

The present study had two aims. First, we sought to identify factors associated with maximum Educational Attainment (EA) in a large sample of adult patients diagnosed with a psychotic disorder in order to clarify the relationship between psychotic illness and EA, as well as to gain insight into specific factors that could plausibly have influenced EA in those individuals. Knowledge of such factors might serve as a foundation for developing educational programs directed toward improving EA in children and adolescents who may be vulnerable to mental illness or who are showing prodromal symptoms of psychotic disorders. A second aim was to examine the relationship between EA and some indicators of economic and social well-being, such as employment and civil status, symptom severity, and alcohol and substance abuse, in order to explore the correlates of low EA in the present patient population.

## 2. Material and Methods

### 2.1. Participants and Study Design

The present dataset was compiled from the medical files of 1132 patients who entered the schizophrenia clinic of the Psychiatric Hospital Fray Bernardino Alvarez in Mexico City from mid-2009 to the end of 2010. This dataset was constructed by psychiatrists from the schizophrenia clinic as part of their normal clinical activities. The primary criterion for inclusion within the dataset was a suspected diagnosis of schizophrenia, for which they were referred to the schizophrenia clinic. At the time of referral, basic clinical data had already been collected by interviewing each patient and a close family member. These data included the age of first psychotic episode, family history of mental illness, and the patient’s current relationship status. At the time of entry into the schizophrenia clinic, all patients were receiving pharmacological treatment and were symptomatically stable. All patients had given their written informed consent for treatment within the hospital.

At the schizophrenia clinic, the patient and family member were further interviewed. The Structured Clinical Interview for DSM-IV (SCID) Axis I Disorders was applied in order to confirm or update the previous DSM-IV diagnosis. The patients received a diagnosis of schizophrenia, bipolar disorder (the majority being bipolar I), schizoaffective disorder, or some other disorder. Positive and negative symptoms in a subset of patients were assessed using the Positive and Negative Syndrome Scale (PANSS). PANSS scores were obtained from a total of 879 patients (780/830 schizophrenia, 11/210 bipolar, 61/78 schizoaffective, 5/9 first episode psychosis, 1 schizophreniform, 1 schizotypical, see Table 1).

Additionally, the patient’s primary caretaker (most often a close family member) was questioned on whether the patient had shown any of the following behavior characteristics prior to the first psychotic symptoms: schizoid-like (isolated, solitary, without friends, did not like to socialize); hyperactive (restless, motor hyperactivity); irritable (angered very easily and frequently); other (e.g., timid, withdrawn, inhibited); or intellectual disability. For the purposes of the present analysis, we considered these to be premorbid behavioral characteristics. Due to the limitations of the interviewing process, we could not distinguish between premorbid personality traits and possible prodromal symptoms. Finally, patients were asked about the maximal level of education that they had reached, with respect to the following categories: (0) no formal education; (1) began grade school but did not finish; (2) finished grade school (grades 1–6); (3) began middle school, but did not finish; (4) finished middle school (grades 7–9); (5) began high school but did not finish; (6) finished high school (grades 10–12); (7) finished some kind of technical career training (most often a substitute for high school); (8) began a university program but did not finish; (9) finished a bachelor’s degree; and (10) finished some postgraduate study. Patients were asked about the maximal level of education according to the above categories, rather than number of years of education in order to avoid confounding results due to cases where a patient may have repeated one or more grade years, thereby adding years of education without advancing with respect to the level of education. Therefore, statistics were performed considering EA as a categorical variable rather than a continuous one (see Section 2.2). Patients’ identities were coded, and all of this information was entered into a database.

The following variables were analyzed in the present study and were classified as either possible ontogenic factors or factors associated with the psychiatric diagnosis or the present state of the patient:

(i) Ontogenic: Family history of mental illness (yes/no), family member with mental illness (none, mother, father, sibling, child, other), familial diagnosis (none, SCH, BD, depression, other), obstetric trauma (yes/no), type of obstetric trauma (none, hypoxia, premature, pre-eclampsia, other), breastfed (yes/no), premorbid characteristics (none, schizoid, irritable, hyperactive, intellectual disability, borderline, withdrawn, obsessive, depressive), family history of metabolic disease (none, diabetic mother, father, both parents, sibling, grandparent, other), sex, birth order, age of father, age of mother, and number of months breastfed.

(ii) Present state: Diagnosis (SCH, BD, schizoaffective), civil status (with partner, no partner), tobacco abuse, excessive consumption of soft drinks, substance abuse, alcoholism, suicide attempt, functional activity (yes/no), type of employment (commercial, construction, cleaning, fieldwork, office, driver, professional, other), independent living (alone/spouse, nonindependent), age of patient, PANSS positive, PANSS negative, PANSS total, number of children, and duration of illness.

### 2.2. Statistical Analysis

Analyses were performed using SPSS software (v.17.0; IBM, www.spss.com). For descriptive statistics, the Student’s *t* test was applied for comparisons of continuous data, and Fisher’s exact test or the Chi2 test were used to compare proportions. 

The Kaplan–Meier survival test was used to analyze the effect of sex on maximum Educational Attainment, where the maximum education level was the time-dependent survival event. Cox regression was applied in order to analyze the contributions of a number of predictor variables of interest on the maximum education level achieved. The hazard event was the level at which the individual terminated formal education. For the first Cox regression model, predictor variables that had been shown in the descriptive analysis to be associated with EA were entered into the model as a single block. This same model was then repeated for male and female patients separately. For the second Cox regression model, predictor variables corresponding to the state of the patient at the time of hospitalization and that had been associated with EA in the descriptive analysis were entered as a single block. This model was then repeated for male and female patients separately. 

## 3. Results

### 3.1. Characteristics of the Patient Sample

The sample comprised 1132 patients who entered Fray Bernardino Psychiatric Hospital between 2010 and 2012 with psychotic symptoms. They were directed to the Schizophrenia Clinic, where patient information was collected, and diagnoses were made. Table 1 shows the general profile of the entire patient population, along with those of the female and male subgroups. SCH was the most common diagnosis, followed by BD and schizoaffective disorder. Female and male patients differed significantly with respect to patient age and the age of symptom onset. SCH occurred more frequently in males than in females, while BD and schizoaffective disorder occurred more frequently in females than in males.

### 3.2. Educational Attainment (EA) of the Patient Sample

Data on the maximum EA are shown in Table 2, expressed as percentages of the total patient population as well as the female and male patient subgroups.

A significantly greater proportion of women than men reported that they did not have formal education beyond primary school, while a greater proportion of men than women reported entering secondary school but not finishing. Similarly, a greater proportion of men than women reported entering high school but not finishing, while a greater proportion of women reported completing high school.

Although the present study did not include a healthy control group, there are publicly available government statistics on maximum EA in the general population of Mexico. Thus, we compared our data to data published by the Mexican Department of Public Education [21] that correspond to the subset of the general population aged 35–44 during the year 2008 [22]. Table 3 shows the levels of EA corresponding to the present sample compared with this segment of the general population. 

Although we did not perform statistical comparisons between these two groups of data, in general, a higher percentage of the present patient sample had completed their basic education and high school compared with the general population.

When we considered the educational categories described above (0 = no formal education; 10 = completed postgraduate study), we found that, for the entire sample (*n* = 1129 cases, 3 missing cases), the mean educational category achieved was 5.05 (SD = 2.43), which corresponds to approximately 9–11 years of formal education. Mexican government statistics for the year 2008 report that the mean number of years of education for the total population aged 35–44 was 8.8 [21,22]. Notably, for the general population, years of education was reported to differ significantly between urban and rural areas (9.9 and 5.8 years, respectively), between nonimpoverished and impoverished populations (10.5 and 5.6 years, respectively), and between nonindigenous and indigenous populations (9.1 and 5.5 years, respectively) [21,22] Thus, although we do not have information on socioeconomic variables for the present population sample, it appears to most closely resemble a generally urban, nonpoor, nonindigenous sample population with respect to the level of EA. This is a reasonable assumption given that the psychiatric hospital is located within the largest urban area of Mexico. 

In order to examine possible differences between male and female patients with respect to EA, we carried out a Kaplan–Meier survival analysis considering the EA categories (0–10, see Methods). For men and women, respectively, the mean survival estimates were 5.097 (SD = 0.089, *n* = 691, 21 cases censored) and 5.138 (SD = 0.129, *n* = 407, 10 cases censored); this difference was not statistically significant (Chi2 = 0.551, df = 1, *p* = 0.458), and was in the direction of favoring women patients. (For reference, category 5 corresponds to beginning high school, but not finishing). In contrast, government statistics from 2008 indicate a small but statistically significant difference between men and women with respect to the mean number of years of education (men, 9.0 years; women, 8.6 years) [21,22].

A second Kaplan–Meier survival analysis was conducted in order to compare the SCH patient group with the patient group diagnosed with some other disorder (BD, schizoaffective disorder, other). For non-SCH and SCH diagnoses, respectively, the mean survival estimates were 5.56 (SD = 0.15, *n* = 302, 16 cases censored) and 4.95 (SD = 0.08, *n* = 812, 15 cases censored); this difference was statistically significant (Chi2 = 14.62, df = 1, *p* < 0.001). Thus, consistent with a number of other published studies, an SCH diagnosis was associated with earlier educational abandonment compared with non-SCH diagnoses (primarily BD and schizoaffective disorder). 

### 3.3. Relationship of EA with Ontogenic Factors

We created a dichotomous outcome variable to represent the educational level achieved, with a 9th grade education or less being considered lower EA and at least some high school education or greater being higher EA. A series of exploratory Student’s *t* tests indicated that the following variables were associated with high EA: a family history of schizophrenia or bipolar disorder, having a mother or second-degree relative with a psychiatric illness, obstetric trauma, and having a grandparent with a metabolic disorder. Interestingly, psychiatric illness of the father or the siblings was not associated with EA. Low EA was associated with having a premorbid intellectual disability or schizoid-like personality characteristics, having a sibling with a metabolic disorder, a longer period of having been breastfed, and a higher birth order (Table 4).

### 3.4. Cox Regression Analysis: Relationship of Ontogenic Factors with Maximum EA

We carried out Cox regression analyses, with abandonment of the educational trajectory being the time-dependent hazard event. The following covariables (which showed statistical significance in the exploratory analysis; Table 4) were entered as a single block: family history of any psychiatric disorder (binary variable), obstetric trauma (binary variable), premorbid personality characteristics (none, schizoid, irritable, hyperactive, learning disability), familial history of metabolic disorders (none, mother, father, both parents, siblings, grandparents, other relative), number of months that the patient had been breast fed as a child (continuous variable), and birth order (continuous variable). The resulting model was significant (Chi2 = 61.07, df = 18, *p* < 0.001; *n* = 569, 20 cases censored, 544 cases with missing values). Significant covariates were a family history of psychiatric illness (B = −0.184, *p* = 0.034, ExpB = 0.832, 95% CI = 0.702–0.986), months of being breastfed (B = 0.12, *p* = 0.011, ExpB = 1.013, 95% CI = 1.003–1.022), birth order (B = 0.064, *p* < 0.001, ExpB = 1.066, 95% CI = 1.027–1.107), and learning disability as a premorbid condition (B = 1.76, *p* < 0.001, ExpB = 5.81, 95% CI = 2.935–11.482). 

This same analysis was then carried out for male and female patients separately. With regard to men, the model (Chi2 = 36.72, df = 17, *p* = 0.004; *n* = 396, 16 cases censored, 302 cases missing values) indicated significant associations with family history of mental illness (B = −0.218, *p* = 0.038, ExpB = 0.804, 95% CI = 0.654–0.988), birth order (B = 0.064, *p* = 0.003, ExpB = 1.066, 95% CI = 1.022–1.113), months of breastfeeding (B = 0.011, *p* = 0.049, ExpB = 1.011, 95% CI = 1.00–1.023), and premorbid intellectual disability (B = 1.66, *p* = 0.005, ExpB = 5.28, 95% CI = 1.641–16.982). With regard to female patients, the model was significant (Chi2 = 27.80, df = 15, *p* = 0.023; *n* = 173, 4 censored cases, 241 cases missing values). Significant covariates were premorbid schizoid-like characteristics (B = 0.413, *p* = 0.028, ExpB = 1.51, 95% CI = 1.045–2.184) and premorbid learning disability (B = 1.78, *p* < 0.001, ExpB = 5.93, 95% CI = 2.44–14.44). Antecedents of metabolic disorders in the grandparents and birth order showed trend-level significance (B = 1.87, *p* = 0.07, ExpB = 6.46, 95% CI = 0.857–48.72; B = 0.07, *p* = 0.09, ExpB = 1.07, 95% CI = 0.99–1.163, respectively).

### 3.5. Relationship of EA with the Current Situation of the Patient

Next, we carried out an exploratory analysis of associations between EA and variables that reflected the patient’s functionality at the time of diagnosis (Table 5).

A SCH diagnosis was associated with low EA, and a BD diagnosis was associated with a trend toward higher EA. High EA was associated with a “continuous” psychotic symptom profile (as opposed to a prodromal symptom profile, first episode of psychosis, or episodic symptom presentation). Low EA was associated with significantly greater PANSS negative symptoms, while there was a trend toward significantly higher PANSS positive symptoms in patients with high EA. High EA was associated with being engaged in some kind of functional occupation (e.g., paid employment, working at home, being enrolled in an educational program). With regard to the type of occupation or employment, low EA was associated with working in the home and janitorial/cleaning employment, while high EA was associated with professional employment. Lower EA was marginally associated with construction employment, substance abuse, and alcoholism, although these associations failed to reach statistical significance. Although the age of illness onset did not differ between patients with low and high EA, the patients’ mean age was greater in those with lower education, as was the duration of their illness. Patients with a lower EA had a greater mean number of children compared to those with a higher education level. 

### 3.6. Cox Regression Analysis: Relationship of the Current Situation of the Patient and Maximal EA

The second Cox regression analysis considered the following covariates: age of patient (continuous variable), number of children (continuous), duration of illness (continuous), diagnosis (SCH, BD, schizoaffective disorder), evolution of illness (binary: episodic versus continuous), relationship status (binary: having or not having a stable long term relationship), substance abuse (binary: yes/no), alcoholism (binary: yes/no), and type of work (salesperson, construction worker, cleaning/janitorial, fieldwork, office work, driver, professional, other). The resulting model was significant (Chi2 = 69.53, df = 18, *p* < 0.001; *n* = 248, 882 cases with missing values, 2 censored cases). Significant covariates were as follows: number of children (B = 0.365, *p* < 0.001, ExpB = 1.44, 9% CI = 1.24–1.68), a bipolar diagnosis relative to a schizophrenia diagnosis (B = −0.67, *p* = 0.001, ExpB = 0.51, 95% CI = 0.357–0.728), having a professional occupation (relative to the category “other”; B = −1.63, *p* < 0.001, ExpB = 0.196, 95% CI = 0.091–0.424), and performing janitorial/cleaning work (B = 0.678, *p* = 0.039, ExpB = 1.97, 95% CI = 1.034–3.754). With regard to male patients, the model was significant (Chi2 = 40.6; df = 18, *p* = 0.002; *n* = 172, 541 cases with missing values, 1 case censored) and contained the following significant covariates: number of children (B = 0.34, *p* = 0.001, ExpB = 1.41, 95% CI = 1.145–1.725), bipolar diagnosis relative to a SCH diagnosis (B = −0.554, *p* = 0.026, ExpB = 0.575, 95% CI = 0.353–0.935), employed in janitorial/cleaning work (B = 1.235, *p* = 0.033, ExpB = 3.439, 95% CI = 1.107–10.686), and employed as a professional (B = −1.343, *p* = 0.008, ExpB = 0.261, 95% CI = 0.096–0.709). For women, the model was significant (Chi2 = 46.37, df = 14, *p* < 0.001, *n* = 76, 341 cases with missing values, 1 censored case) and contained the following significant covariates: number of children (B = 0.569, *p* < 0.001, ExpB = 1.767, 95% CI = 0.943–1.035), having a bipolar diagnosis relative to a SCH diagnosis (B = −0.853, *p* = 0.01, ExpB = 0.426, 95% CI = 0.223–0.816), being employed in janitorial/cleaning work (B = 1.058, *p* = 0.027, ExpB = 2.88, 95% CI = 1.125–7.382), and being employed as a professional (B = −2.416, *p* < 0.001, ExpB = 0.089, 95% CI = 0.023–0.346).

### 3.7. Independent Associations of Familial History and Diagnosis 

It was of interest to determine whether family history of psychiatric illness (binary; no family history and history of any psychiatric disorder) and diagnosis (binary; nonschizophrenia and schizophrenia) were each independently associated with EA. We therefore carried out a third Cox regression analysis to examine these two covariates specifically. The resulting model was significant (Chi2 = 14.055, df = 2, *p* < 0.001, *n* = 1079, 28 cases censored, 25 cases with missing data). Each covariate was statistically significant, with family history of any psychiatric disorder decreasing the hazard (B = −0.137, *p* = 0.025, ExpB = 0.872, 95% CI = 0.774–0.983) and a schizophrenia diagnosis increasing the hazard (B = 0.201, *p* = 0.004, ExpB = 1.223, 95% CI = 1.068–1.401). When men (*n* = 677, 20 cases censored, 17 cases with missing data) and women (*n* = 402, 8 cases censored, 8 cases with missing data) were analyzed separately, these associations were significant for men only (family history B = −0.162, ExpB = 0.85, *p* = 0.037, 95% CI = 0.731–0.990; schizophrenia diagnosis B = 0.280, ExpB = 1.324, *p* = 0.005, 95%, CI = 1.087–1.612), while not reaching statistical significance in women (family history ExpB = 0.886, *p* = 0.232; schizophrenia diagnosis ExpB = 1.126, *p* = 0.25).

## 4. Discussion

In the present study, we found a number of factors that were significantly associated with EA in a large sample of patients who had sought treatment for psychotic symptoms. These factors are summarized in Table 6.

Some were premorbid or ontogenic factors (i.e., factors present before the onset of psychotic symptoms) that could reasonably have influenced the patients’ level of EA, including a family history of mental illness (which predicted higher EA), an intellectual disability, premorbid schizoid-like personality characteristics, months that the patient was breastfed as an infant, and birth order (which all were associated with lower EA). Our data also suggest the involvement of gender-specific differences in the contributions of these factors to EA: having premorbid schizoid-like characteristics was significantly negatively associated with EA in women only, while family history of mental illness had a significant positive association with EA in men only. A second set of associated factors may have been a consequence of lower EA and/or of the mental illness itself. These included the number of children that the patient had and being employed in manual labor. Higher EA was associated with having a bipolar diagnosis (relative to a SCH diagnosis) and having a professional occupation.

Perhaps surprisingly, we found that familial history of mental illness (either BD or SCH) was associated with higher EA within this patient sample. This finding is consistent with a number of recently published reports [10,11,12,14,15,16]. It has been suggested that certain schizotypical traits associated with vulnerability to psychotic disorders may confer greater creativity and, as a consequence, favor higher EA [23]. Thus, a diagnosis of BD, as well as a familial history of this disorder, was positively associated with creativity [24,25] Kyaga and colleagues [26] studied creativity in various professions and found that the nonaffected first-degree relatives of patients with BD demonstrated increased creativity compared with a control group. Vreeker and colleagues [18] observed that, although patients with BD had lower mean IQs than healthy controls, they were nevertheless more likely to complete more years of education. Notably, in the present study, we found that a SCH diagnosis (relative to BD diagnosis) and familial history of psychiatric disorders were independently associated with low and high EA, respectively.

Consistent with the association of a SCH diagnosis with low EA, we found a negative association between premorbid schizoid-like characteristics and EA. Schizotypy, as a psychological construct, is defined as a collection of personality traits believed to be related to psychosis risk, including dysfunctional social interactions, reduced social competence, social anhedonia, reduced facial, vocal, and gestural expressions, low positive affect, and high negative affect [27]. In the context of psychotic disorders, deficient social interactions may be related to specific cognitive deficits, including the inability to perceive facial emotion expressions and vocal prosody [27]. In general, schizotypy is more common in males; indeed, in the present patient sample, men were 1.7 times more likely than women to have exhibited premorbid schizoid personality characteristics (data not shown).

Our results indicate that having exhibited premorbid schizoid-like characteristics is significantly associated with lower EA in female patients but not in males. Although schizotypy was not formally assessed in the patient sample, patients that were qualitatively described as being solitary, isolated, having few friends, and preferring to be alone prior to the onset of psychotic symptoms were considered to have exhibited premorbid schizoid characteristics. It is possible that the association between such characteristics and lower EA could be due to the negative effects of poor social skills during the years of primary and secondary school. Indeed, the impact of psychiatric symptoms on social functioning has been identified as a possible mediating factor for low EA [28]. However, it is difficult to explain how such a deficit could affect girls more than boys. Nevertheless, such children might constitute an important target population for mitigating the effects of mental illness on EA and later functionality [1]. These children could be channeled into specially designed programs that are aimed at improving social skills and providing the necessary guidance and support to encourage the child to complete formal education. Alternatively, home or individualized schooling might also be useful options for these children. 

The association between premorbid schizoid characteristics and lower EA in our sample emphasizes the importance of considering mental health when formulating educational policies aimed at reducing school dropout rates. A diagnosis of severe mental illness (including SCH, BD, and major depression) before the age of 25 has been associated with economic burden across the lifespan, impacting employment, lifetime earnings, and economic stability [1]. Our results suggest that some premorbid characteristics or prodromal symptoms that precede a SCH diagnosis may negatively impact EA, which in turn may impact the functionality of the adult patient. Indeed, low EA in our sample was associated with manual labor (typically with lower pay and benefits compared to professional occupations) and a greater number of children (which contribute to the patient´s economic burden).

The present data indicate that higher EA is associated with continuous psychotic symptoms and higher PANSS positive scores. In contrast, lower EA is associated with higher PANSS negative scores and premorbid schizoid-like characteristics. Taken together, these results suggest that EA is negatively impacted by the severity of negative symptoms (reflected by PANSS negative scores and premorbid schizoid characteristics), while possibly being unaffected or positively impacted by (genetic?) factors associated with the severity of positive symptoms (reflected by higher PANS positive scores and a continuous symptom profile). In support of this proposal, negative symptoms (along with decreased cognitive functioning and lower education level), but not positive symptoms, were found to be a negative predictor of employment in a meta-analysis [29]. In contrast, positive schizotypy scores (the dimension of “unusual experiences”) but not negative schizotypy scores (introvertive anhedonia) were associated with high creativity [23]. In the present patient sample, these factors must have exerted their impacts before the manifestation of overt psychiatric symptoms, as the mean age of symptom onset was approximately 25 years (well after high school), and this did not differ between patients with low and high EA. In the case of patients with elevated negative symptoms, the premorbid presence of schizoid-like characteristics could have favored dropping out of school during the early school years, as discussed above. Indeed, the subset of psychotic patients who reported premorbid schizoid-like characteristics showed significantly higher PANSS negative scores (but not PANSS positive scores) than those who did not, consistent with the proposal that premorbid schizoid-like characteristics may be predictive of a higher negative symptom SCH profile (data not shown). In the case of patients with elevated positive symptoms, genetic factors that promote creativity could have favored higher EA. An alternative possibility is that the cognitive and social demands of higher education might have triggered the onset of psychotic symptoms in genetically predisposed individuals who had already reached higher levels of education. 

A report from The National Institute of Education (Mexico) [21] identified some principal reasons for dropping out of school, including a lack of money, not enjoying study, or considering it more important to work. In women specifically, principal reasons included a lack of money, becoming pregnant, or getting married [21]. In this context, it is notable that, in the present study, lower EA was associated with a greater number of children in both male and female patients. Nevertheless, female patients reported being a parent to approximately twice the number of children as men (data not shown), a gender effect also reported by Bhatia and colleagues [30]. Moreover, the negative effect on EA of having children was greater for women than for men (see Table 6). Although we do not have data on the age at which these patients first became parents, it is clear that pregnancy could have had a severe impact on the capacity of a young woman to continue her formal education. Young mothers tend to prematurely terminate their education, as has been described for young Mexican American mothers [31]. Moreover, girls commonly have more demands from their family and fewer opportunities than boys; such stressors associated with the role of caregiver, mother, or head of the family could further negatively impact mental health and functionality [32]. 

In the present study, as in a previous analysis of the same patient population [33], there were notable differences between male and female patients with regard to several measures (for a review of gender differences in SCH see Ochoa et al. 2012 [34]). Notably, men were twice as likely than women to report having dropped out of school at the secondary or high school levels. It seems likely that this difference was due, at least in part, to prodromal characteristics specific to SCH (the most common diagnosis in men), such as cognitive deficits. Indeed, our analyses indicate that a BD diagnosis is associated with 57% and 42% decreases in the hazard (compared to having a SCH diagnosis) for dropping out of school in women and men, respectively. Premorbid schizoid-like characteristics had a statistically significant negative impact on EA in women only, while negative effects of birth order and duration of breastfeeding were significant only in men. We suggest that these sex-specific negative effects may be mediated by social and economic factors: (i) since schizoid-like characteristics are more common in boys in general, such characteristics might be more socially acceptable in boys than in girls, perhaps making it more difficult for girls with such characteristics to function within the typical childhood social milieu; and (ii) higher birth order and longer duration of having been breastfed might be associated with a lower socioeconomic status. Interestingly, a family history of mental disorder was significantly associated with higher EA in men only. This result coincides with a recent study by Velthorst et al. [35], which found that siblings of patients with SCH outperformed control subjects with respect to certain cognitive measures (visuospatial and processing speed) after adjusting for symptom severity and functioning. Moreover, male siblings of SCH patients were found to have better cognitive function than female siblings, indicating a sex-specific genetic risk factor associated with both SCH and improved cognition.

Although we did not have a nonpsychiatric control group with which to compare our patient sample, there are publicly available governmental statistics that were obtained for an equivalent age group and corresponding to the year 2008, which represents a time frame similar to that during which our data were collected (2010–2012). When we compared these data (corresponding to the general population of Mexico) with our patient population, we observed that, in general, our patient population had slightly higher EA than the general population—in particular, at the high school level. This finding is consistent with the finding that the educational gap between patients with SCH and healthy control individuals is lower in lower-income countries compared with high-income countries [15]. Additionally, although the national statistics of the general population show a significant difference between women and men with respect to EA (women having a significantly lower EA), men and women in our patient sample did not show such a difference. Differences between our patient sample and the general population with respect to EA could have been due to the environmental and socioeconomic characteristics of our sample (e.g., urban versus rural living, nonimpoverished versus poverished, nonindigenous versus indigenous). Although we do not have such information for our patient population, given that the psychiatric hospital is in a major urban area, we assume that the patient population was largely urban and of mixed ancestry.

There are a number of important limitations to the present study. Thus, the study was retrospective and based largely on information that was obtained by interviewing the patient and/or caregiver and which could not be objectively confirmed. Perhaps most notably, it was not possible to verify the patients’ statements about the maximal level of education that they had reached. Moreover, since the study lacked a healthy control comparison group, our data cannot speak to the possible relationships among various factors (e.g., family history of mental illness, schizoid-like personality characteristics, birth order) and EA in the nonclinical population. Finally, we acknowledge that the data were collected over 10 years ago and may not necessarily fully represent the current situation. However, we argue that the relationships between EA and specific diagnoses, family history of mental illness, sex, premorbid personality characteristics, and other factors that were examined here seem unlikely to have changed significantly during this time: the fundamental characteristics of psychotic disorders clearly have not changed, nor has the manner in which these diseases are treated, nor has the educational system.

## 5. Conclusions

The present results are in agreement with a large body of studies showing that, in general, individuals with SCH often suffer from academic underperformance prior to illness onset (in primary, secondary, and high school) and are less likely to have completed secondary and high school compared with healthy controls or, as in the present case, compared with individuals having some other non-SCH diagnosis. They are also consistent with a complementary body of literature that argues that psychotic disorders—or genetic vulnerability to them—confer a certain academic advantage or capacity for higher creativity. We conclude that specific diagnosis and genetic vulnerability to psychotic disorders are each independently associated with EA: while certain prodromal characteristics of SCH—such as schizoid-like personality characteristics—may negatively impact early academic performance, individuals with genetic vulnerability to BD (and to a lesser extent to SCH) may have some academic advantage over those that have no family history of mental illness. The present results help to clarify the seemingly contradictory conclusions of the published literature in this area (e.g., see Introduction).

In Mexico, there is a scarcity of published information about the impact of mental illness on EA; therefore, the present study is useful in the context of understanding the implications of psychotic disorders on EA in Mexico as well as in other countries with similar socioeconomic profiles. Childhood and adolescence represents a critical window of time in which some of the later functional impairments of major mental illness might be prevented, and improving EA could mitigate some of the negative economic and social impacts of SCH and other severe mental illnesses. In our sample population, lower EA was associated with having more children and employment involving manual labor. Given that major mental illness has been associated with significant direct and indirect costs and that increased EA might mitigate some negative economic effects of mental illness [1], identifying children at high risk for SCH—in particular, girls who display schizoid—like characteristics—and channeling them into programs and counseling aimed at keeping them in school could have a significant impact on reducing the individual and social economic burdens of SCH in the event that these children later develop this illness.

## Figures and Tables

**Table 1 brainsci-13-00881-t001:** Clinical characteristics of the patient sample. Data are presented as the mean (SD) or frequency (percent of sample).

	All Patients (*n* = 1132)	Female (*n* = 418)	Male (*n* = 714)	Test Statistic
Age (years)	36.44 (10.54)	38.14 (10.55)	35.44 (10.41)	***t*(1130) = 4.19, *p* < 0.001**
Age of first psychotic episode	24.75 (8.62)	26.89 (9.38)	23.51 (7.99)	***t*(73) = 6.13, *p* < 0.001**
Duration of illness (years)	11.59 (9.27)	11.01 (8.94)	11.92 (9.44)	NS
Schizophrenia (SCH)	830 (73.3%)	247 (59.1%)	583 (81.7%)	***p* < 0.0001**
Bipolar disorder (BD)	210 (18.6%)	130 (31.1%)	80 (11.2%)	***p* < 0.0001**
Schizoaffective disorder	78 (6.89%)	38 (9.1%)	40 (5.6%)	***p* = 0.029**
First psychotic episode	9 (0.79%)	3 (0.7%)	6 (0.8%)	NS

Notes: Bold type denotes a significant difference between female and male patients (*t*-test or Fisher’s Exact Test), NS denotes a nonsignificant comparison.

**Table 2 brainsci-13-00881-t002:** Maximal Educational Attainment reported by the patient sample. Data are presented as the frequency (percent of sample).

Maximal Educational Attainment (EA)	Total Sample (*n* = 1129)	Female (*n* = 417)	Male (*n* = 712)	Test Statistic
None	10 (0.9%)	5 (1.2%)	5 (0.7%)	NS
Primary incomplete	70 (6.2%)	32 (7.7%)	38 (5.3%)	NS
Primary complete	**124 (11%)**	**60 (14.4%)**	**64 (9.0%)**	***p* = 0.006**
Secondary incomplete	**78 (6.9%)**	**18 (4.3%)**	**60 (8.4%)**	***p* = 0.008**
Secondary complete	238 (21%)	81 (19.4%)	157 (22%)	NS
High school incomplete	**168 (14.8%)**	**34 (8.1%)**	**134 (18.8%)**	***p* < 0.0001**
High school complete	**82 (7.2%)**	**58 (13.9%)**	**24 (3.4%)**	***p* < 0.0001**
Technical school	111 (9.8%)	38 (9.1%)	73 (10.2%)	NS
Bachelor’s incomplete	122 (10.8%)	40 (9.6%)	82 (11.5%)	NS
Bachelor’s complete	118 (10.4%)	46 (11%)	72 (10.1%)	NS
Master’s and above	8 (0.7%)	5 (1.2%)	3 (0.3%)	NS

Bold type denotes significant difference between female and male patients (Fisher’s exact test), NS denotes a nonsignificant comparison.

**Table 3 brainsci-13-00881-t003:** Comparison of maximal Educational Attainment of the present patient sample (mean age = 36 years) to that of the general population aged 35–44 for the year 2008 (figures obtained from Mexican Department of Education publications).

Educational Attainment	Present Sample	General Population Aged 35–44
None	0.9%	4.9%
Incomplete basic education	24%	36.1%
Completed basic education (primary and secondary levels)	36%	34.2%
Completed high school or technical training	27.8%	13.1%
Completed university education	11.5%	11.8%

**Table 4 brainsci-13-00881-t004:** Clinical characteristics that differed significantly between patient groups with low or high Educational Attainment. Data are expressed as the frequency (percent) or mean (SD); Fisher’s exact and t statistics are shown.

	Low EA	High EA	Test Statistic
Familial SCH	160/506 (31.6%)	219/593 (36.9%)	*p* = 0.065
Familial BD	31/506 (6.1%)	66/593 (11.1%)	*p* = 0.004
Mental illness of mother	23/509 (4.5%)	45/595 (7.6%)	*p* = 0.044
Mental illness of second-degree relative	81/509 (15.9%)	131/595 (22%)	*p* = 0.011
Obstetric Trauma	72/285 (25.3%)	119/362 (32.9%)	*p* = 0.037
Premorbid intellectual disability	10/358 (2.8%)	0/436 (0%)	*p* = 0.0003
Premorbid schizoid	166/358 (46.4%)	152/436 (34.9%)	*p* = 0.001
Metabolic disorder: grandparent	36/507 (7.1%)	80/598 (13.4%)	*p* = 0.0008
Metabolic disorder: sibling	36/507 (7.1%)	18/598 (3%)	*p* = 0.002
Months breastfed	10.69 (10.07)	6.87 (7.27)	*t*(472) = 5.25, *p* < 0.001
Birth order	3.94 (2.71)	3.10 (2.25)	*t*(986) = 5.49, *p* < 0.001

**Table 5 brainsci-13-00881-t005:** Characteristics of patient groups with low or high Educational Attainment.

	Low EA	High EA	Test Statistic (Fisher’s Exact Test; Student’s *t*-Test)
SCH diagnosis	**402/522 (77%)**	**425/607 (70%)**	***p* = 0.009**
BD diagnosis	85/522 (16.3%)	125/607 (20.6%)	*p* = 0.066
Schizoaffective diagnosis	33/522 (6.3%)	45/607 (7.4%)	*p* = 0.483
Continuous symptom profile	**49/520 (9.4%)**	**84/605 (13.9%)**	***p* = 0.021**
Engaged in functional activity	**217/354 (46.7%)**	**253/404 (62.6%)**	***p* < 0.0001**
Paid work	121/354 (34.2%)	163/404 (40.3%)	*p* = 0.084
Working at home	**83/353 (23.4%)**	**61/404 (15.1%)**	***p* = 0.004**
Construction	11/118 (9.3)	6/159 (3.8%)	*p* = 0.076
Janitorial/cleaning	**11/118 (9.3%)**	**4/159 (2.5%)**	***p* = 0.016**
Fieldwork	1/118 (0.8%)	2/159 (1.3%)	NS
Professional occupation	**0/118**	**12/159 (7.5%)**	***p* = 0.002**
Alcoholism	40/332 (12%)	32/399 (8%)	*p* = 0.081
Substance abuse	62/389 (15.9%)	53/460 (11.5%)	*p* = 0.07
Number of children	**0.94 (1.36) *n* = 360**	**0.46 (0.99) *n* = 416**	***t*(646) = 5.55, *p* < 0.001**
PANSS negative	**3.09 (0.89), *n* = 407**	**2.96 (0.95), *n* = 452**	***t*(857) = 2.00, *p* = 0.045**
PANSS positive	2.62 (1.02), *n* = 407	2.76 (1.07), *n* = 453	*t*(858) = 1.93, *p* = 0.054
Age of first psychotic episode	24.94 (9.1), *n* = 517	24.59 (8.21), *n* = 597	*t*(1048) = 0.671, *p* = 0.50

Note: Data are expressed as the frequency (percent) or mean (SD); Fisher’s exact and t statistics comparing low and high Educational Attainment groups are shown, and significant comparisons are presented in bold type.

**Table 6 brainsci-13-00881-t006:** Summary of results from the Cox regression analysis.

Factor	All Patients	Women	Men
Premorbid Learning Disability	5.81 (+481%)	5.93 (+493%)	5.28 (+428%)
Cleaning/Janitorial Employment	3.75 (+275%)	2.88 (+188%)	3.44 (+244%)
Professional Employment	0.196 (−80.4%)	0.089 (−91%)	0.26 (−74%)
Premorbid Schizoid Characteristics	NA	1.51 (+51%)	NA
Bipolar Diagnosis	0.51 (−49%)	0.426 (−57.4%)	0.575 (−42%)
Number of Children	1.44 (+44%)	1.77 (+77%)	1.14 (+14%)
Family History of Mental Illness	0.832 (−16.8%)	NA	0.804 (−19.6%)
Birth Order	1.066 (+6.6%)	NA	1.066 (+6.6%)
Months Breastfed	1.013 (+1.3%)	NA	1.011 (+1.1%)

Note: Factors that were found to be associated with the maximal level of education (i.e., level at which education was terminated, time-dependent variable). Hazard ratios (ExpB) are shown, along with the corresponding percent increase/decrease in the hazard for terminating education. Factors are ordered from the largest to the smallest effect size.

## Data Availability

Data are available on request from the corresponding author.

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
