# Peer review of "Analysis of Educational Attainment in a Mexican Psychiatric Patient Population with Bipolar or Psychotic Disorders"

_brainsci, 2023, doi:10.3390/brainsci13060881_

Round 1

Reviewer 1 Report

The aim of the study is not clear. An option may be the mention the level of education among pts diagnosed with schizophrenia, bipolar disorder or schizoaffective disorder. In such a case the differention between pts with various diagnoses should be performed. And it has't been done.

Another option is an attempt to indicate which of the factors studied may influence the presence of various diagnoses and may be associated with the severity of symptoms. In such a case the length of treatment should be taken into account. The third option is a comparison of eductional levlels between pts with mental disorders with control group. This has been done but -at less in a limited extend.

In summary the aim of the study should be clearly stated.

Another problem is that it is hard to understand whether the authors have discovered what is the influence of low or high EA on the incidence and the course of mental disorders? Clinical experience and common sense says that pts with psychotic disorders in general are less educated. Did the authors confirm tha conclusion or not? If not, what explanation are they supposed to give?

Why the duration of untreated psychosis was not taken into consideration? The age of onset and the DUP are different factors.

Author Response

REVIEWER 1

Comments and Suggestions for Authors

The aim of the study is not clear. An option may be the mention the level of education among pts diagnosed with schizophrenia, bipolar disorder or schizoaffective disorder. In such a case the differention between pts with various diagnoses should be performed. And it has't been done.

Another option is an attempt to indicate which of the factors studied may influence the presence of various diagnoses and may be associated with the severity of symptoms. In such a case the length of treatment should be taken into account. The third option is a comparison of eductional levlels between pts with mental disorders with control group. This has been done but -at less in a limited extend.

In summary the aim of the study should be clearly stated.

RESPONSE: The Introduction section has been extensively reworked in order to more clearly communicate the principal aims of the study and their relevance: 1) to identify factors associated with low (and high) educational attainment in this patient sample, and 2) examine the relationship between maximal EA and some indicators of economic and social well-being. With respect to the question of whether EA differed between SCH and non-SCH diagnoses, the revised version includes a Kaplain-Meier survival analysis that addresses this question (please see lines 235 - 243).

Another problem is that it is hard to understand whether the authors have discovered what is the influence of low or high EA on the incidence and the course of mental disorders? Clinical experience and common sense says that pts with psychotic disorders in general are less educated. Did the authors confirm tha conclusion or not? If not, what explanation are they supposed to give?

RESPONSE: Firstly, we acknowlege that in a cross-sectional study such as this one, cause-effect relationships cannot be established. Secondly, we would like to emphasize that the relationship between EA and psychotic disorders (specifically, SCH and BD) involves a number of intermediary factors, and has not always been found to be negative (see Introduction). In the present analyses, we identified statistical associations between EA and a number of factors that had previously been suggested to increase risk for psychotic disorders (sex, familial history of mental disorder, obstetric trauma, premorbid personality characteristics, etc.), in order to determine whether any of these might also be associated with EA, independently of the patient's diagnosis. Since these factors were present during the educational trajectory of the patient (before onset of psychotic symptoms), any of them could have plausibly influenced maximal EA. We also identified associations between EA and various indirect indicators of present function (civil status, employment, diagnosis, symptom severity, etc.). We found that, within this patient population, a SCH diagnosis was associated with lower EA, while a BD diagnosis was associated with higher EA (e.g., lines 235 - 243). We also found that specific diagnosis and familial history of a mental disorder each were independently associated with EA: SCH diagnosis was associated with lower EA (relative to patients with a BD or schizoaffective disorder diagnosis), while familial history of a mental disorder was associated with higher EA relative to patients with no family history) (lines 343 - 359). Our data also suggest that more prominent negative symptoms were associated with low EA, while more prominent positive symptoms were associated with higher EA (discussed in lines 439 - 467). Taken together, the present results suggest that EA in patients diagnosed with psychotic disorders may be influenced by a number of factors, including familial history of mental disorders and premorbid personality characteristics.

Since a schizophrenia diagnosis was associated with low EA, yet overt symptoms of schizophrenia were not present during the patient´s educational trajectory, we can speculate that prodromal symptoms or characteristics specific to schizophrenia (e.g., cognitive deficits, premorbid schizoid-like characteristics) may have negatively influenced EA in this patient population. By contrast, familial history of mental disorders (genetic liability) was positively associated with EA, independently of diagnosis. This latter result is consistent with a number of genetic studies, as well as with studies that have argued that schizophrenia is associated with higher creativity (the results of these studies were summarized in the Introduction).

Finally, it is important to emphasize that, since we do not have a healthy control group for comparison, we cannot formally determine whether this patient population (or individual diagnoses within this population) had lower EA than healthy subjects. However, given the information available on EA in the general Mexican population at around the time of data collection, the patient sample is on par with or slightly higher than the general population with respect to EA (discussed in lines 512 - 528). We suggest that this result might be due to socioeconomic differences between the patient population and the general population: the patient population is likely over-represented by persons from the metropolitan area of Mexico City, while a significant proportion of the general population lives in rural areas.

Why the duration of untreated psychosis was not taken into consideration? The age of onset and the DUP are different factors.

RESPONSE: Unfortunately, we do not have data on duration of untreated psychosis. However, since the mean age of the first psychotic episode (around 25 years old) was well after high school should already have been completed, it is unlikely that DUP could have impacted on EA. On the other hand, DUP could have impacted on the indicators of functionality (employment, civil status, etc.). Nevertheless, because we lack information on DUP, we cannot investigate these possibilities.

Reviewer 2 Report

The thematic issue of the paper is quite interesting and the large sample presented is very rare to accomplish in this kind of studies. Clinical samples are very rare with all pathologies, in particular with this one, the high number of patients is very useful. The conclusions are also very useful to the practitioners and researchers in the brain sciences field. Highly recommend the publication

Author Response

REVIEWER 2

Comments and Suggestions for Authors

The thematic issue of the paper is quite interesting and the large sample presented is very rare to accomplish in this kind of studies. Clinical samples are very rare with all pathologies, in particular with this one, the high number of patients is very useful. The conclusions are also very useful to the practitioners and researchers in the brain sciences field. Highly recommend the publication

RESPONSE: We thank the reviewer for their positive feedback.

Reviewer 3 Report

1.Author affiliation information should be in English.

2.Quality of Presentation is very bad.

3.Extensive editing of English language and style required.

1. The novelty of the study was insufficient,Data collection was carried out from mid-2009 until the end of 2010, More than 10 years ago, it is difficult to reflect the current state of education of people with schizophrenia in Mexico?
2. Data on educational attainment in healthy controls should use data on the educational attainment of men and women in healthy Mexican populations for the same period 2009–2010.
The data collected by the study is now more than 13 years old.

Author Response

REVIEWER 3

1.Author affiliation information should be in English.

RESPONSE: We have made this correction.

2.Quality of Presentation is very bad.

RESPONSE: We are unsure of exactly to which aspects of the presentation the reviewer is referring. However, in response to the comments of other reviewers, we have restructured the Introduction and Discussion sections; we hope these changes address the concerns of this reviewer.

3.Extensive editing of English language and style required.

RESPONSE: The corresponding author, Kurt Hoffman, is a native English speaker and an experienced author of many published works written in English. Therefore, we are confident that the manuscript is free of grammatical errors. With regards to style: in response to the comments of other reviewers, we have restructured the Introduction and Discussion sections; we hope these changes address the concerns of this reviewer.

The novelty of the study was insufficient,Data collection was carried out from mid-2009 until the end of 2010, More than 10 years ago, it is difficult to reflect the current state of education of people with schizophrenia in Mexico?

RESPONSE: The reviewer's concern on the age of the data is well taken. In the revised version, We acknowledge that an important limitation of this study is that the data were collected more than 10 years ago (lines 535 - 542). However, we would argue that the relationships between EA and specific diagnoses, family history of mental illness, sex, premorbid personality characteristics, and other factors that were examined here seem unlikely to have changed significantly during this time.

Regarding the novelty of the study: We believe that our study contributes important information on factors that may influence EA in individuals with high risk for psychotic disorders: sex, family history of mental disorders, prodromal symptoms or certain premorbid personality characteristics. This information can be taken into account in order to identify students at higher risk for abandoning the educational trajectory, and steps could be taken to prevent them from doing so. For example, children and adolescents – particularly girls – that show schizoid-like behavioral traits (poor social skills, low social motivation) might be channeled into specialized therapeutic programs or more individualized education.

Data on educational attainment in healthy controls should use data on the educational attainment of men and women in healthy Mexican populations for the same period 2009–2010.
The data collected by the study is now more than 13 years old.

RESPONSE: We acknowledge that an important limitation of the present study is the lack of a healthy control group (see lines 532 - 526) . However, we did have access to statistics (census data) on EA of the general population of Mexico for an equivalent age group, for the year 2008. We did not consider it valid to statistically compare data from our patient population with data from the general population. However, we present these data in Table 3 as well as in the text (lines 197 – 224; 512 - 531). On the whole, EA in the patient population was quite similar to that of the general population. Differences between the patient population and the general population were most likely due to socioeconomic differences, as discussed in the text.

Reviewer 4 Report

This is an interesting paper investigating the educational attainment in patients suffering from psychosis and bipolar disorders. The paper is well written and of interest for the readers; however, I recommend several changes to improve the paper.

ABSTRACT.

1- There is no abstract in the main document before the introduction section. I recommend to add it.

INTRODUCTION

1- The first part of the introduction is mainly focused on psychotic symptoms. It describes its presence in patients with bipolar disorder and schizophrenia, and afterwards, they describe the shared genetic vulnerability. I recommend to start with the concept of severe mental illness. It seems more indicated to introduce it before describing several diseases. 

2- Lines 105-113. At the end of the introduction, the authors report what they are examining. I recommend to build a subsection called "1.1. Aims", and to describe more clearly the main aims of that paper.

MATERIAL AND METHODS

1- I recommend to rename the section as "2.1. Participants and study design". 

2- Table 1 should be removed from the methods section and presented in the results section.

RESULTS

1- The first part of the results should be the description of the main characteristics of the sample. I recommend to summarize them and rename the section.

2- High school completed is more prevalent in women. This results is very interesting and should be discussed in terms of gender differences in psychosis and schizophrenia. There are several papers from Prof. Seeman and Prof. Riecher-Rössler about this field. I recommend to discuss them in the discussion section.

CONCLUSIONS

- A conclusions section is needed.

Author Response

REVIEWER 4

This is an interesting paper investigating the educational attainment in patients suffering from psychosis and bipolar disorders. The paper is well written and of interest for the readers; however, I recommend several changes to improve the paper.

ABSTRACT.

1- There is no abstract in the main document before the introduction section. I recommend to add it.

RESPONSE: We have added an abstract to the manuscript.

INTRODUCTION

1- The first part of the introduction is mainly focused on psychotic symptoms. It describes its presence in patients with bipolar disorder and schizophrenia, and afterwards, they describe the shared genetic

vulnerability. I recommend to start with the concept of severe mental illness. It seems more indicated to introduce it before describing several diseases.

RESPONSE: Thank you for this suggestion. We have re-organized and “streamlined” the content of the Introduction. 

2- Lines 105-113. At the end of the introduction, the authors report what they are examining. I recommend to build a subsection called "1.1. Aims", and to describe more clearly the main aims of that paper.

RESPONSE: We have added the requested subsection in order to more explicitly state the aims of the study.

MATERIAL AND METHODS

1- I recommend to rename the section as "2.1. Participants and study design". 

RESPONSE: We have renamed this section as suggested.

2- Table 1 should be removed from the methods section and presented in the results section.

RESPONSE: The first text reference to Table 1 has now been moved to the Results section, we leave it to the type-setting editors to move the table to the Results section accordingly.

RESULTS

1- The first part of the results should be the description of the main characteristics of the sample. I recommend to summarize them and rename the section.

RESPONSE: We have added this subsection as requested.

2- High school completed is more prevalent in women. This results is very interesting and should be discussed in terms of gender differences in psychosis and schizophrenia. There are several papers from Prof. Seeman and Prof. Riecher-Rössler about this field. I recommend to discuss them in the discussion section.

RESPONSE:

We have added a discussion of sex-dependent differences that we observed in our sample (lines 489 – 514).

CONCLUSIONS

  • A conclusions section is needed.

RESPONSE: We have added a conclusions section.

Round 2

Reviewer 3 Report

The revised version has made significant progress compared to the previous version.

Author Response

We thank this reviewer and all others for their comments and observations on the original version of the manuscript.  We consider that the revised version is much improved as a result of their efforts.